# Self-Reported Allergic Adverse Events Following Inactivated SARS-CoV-2 Vaccine (TURKOVAC™) among General and High-Risk Population

**DOI:** 10.3390/vaccines11020437

**Published:** 2023-02-14

**Authors:** Ateş Kara, Aslihan Coskun, Fehminaz Temel, Pervin Özelci, Selmur Topal, İhsan Ates

**Affiliations:** 1Department of Pediatrics, Pediatric Infectious Disease Unit, Hacettepe University Faculty of Medicine, Health Institutes of Türkiye, Türkiye Vaccine Institute, Ankara 06100, Türkiye; 2Health Institutes of Türkiye, Türkiye Vaccine Institute, Ankara 06260, Türkiye; 3Türkiye Ministry of Health, General Directorate of Public Health, Department of Communicable Diseases and Early Warning, Field Epidemiology Unit, Ankara 06430, Türkiye; 4University of Health Science, Ankara City Hospital, Department of Internal Medicine, Ankara 06610, Türkiye

**Keywords:** COVID-19, vaccine, allergy, adverse events

## Abstract

TURKOVAC™ is a whole-virion inactivated COVID-19 vaccine which was developed and granted emergency use and conditional marketing authorization in December 2021 in Türkiye. The objective of this study is to assess the distribution and the severity of allergic adverse events following the administration of the vaccine as the primary or the booster dose in 15 provinces in Türkiye. In this cohort study, between February and May 2022, in the selected 15 provinces having an adequate number of health care personnel in the community health centers to conduct the study, 32,300 people having the first, the second, or the booster dose of the vaccine were invited to the survey. A total of 29,584 people voluntarily agreed to participate to the survey and were given a structured questionnaire after a minimum of 10 days following the vaccination. In our study, only 0.5% of the participants (142 persons) reported to experience any allergic reaction, and 12 of them (8.5%) reported to be given medical treatment in a health center. Male predominance (55.6%) was observed among participants reported to experience any allergic reaction. No hospitalization was recorded. Of the participants, 4.4% (1315 people) reported to have a history of allergy. The most reported allergens were drugs. Among the participants without a known history of allergy (*n* = 28,269), 0.4% of them (110 people) reported to experience an allergic reaction following the vaccination, and 5.4% of the allergic reactions (six people) were reported to be treated in a health center. The percentage of the participants given any medical treatment among the participants without a known history of allergy is 0.02%. No immediate or anaphylactic reaction was reported. Among the participants with a known history of allergy (*n* = 1315), 32 people (2.4% of them) reported to experience an allergic reaction following the vaccination, and 18.7% of the allergic reactions (six people) were reported to be prescribed a medical treatment. The percentage of the participants given any medical treatment among the participants with a known history of allergy is 0.4%**.** A known history of allergy increased the risk of having an allergic experience by approximately six times following vaccination. As a whole-virion inactivated SARS-CoV-2 vaccine, the TURKOVAC™ vaccine, with a low allergic reaction-related adverse event profile, can be an alternative to other COVID-19 vaccines.

## 1. Introduction

Severe Acute Respiratory Syndrome Coronavirus 2 (SARS-CoV-2) was first detected in Wuhan, China, in late December 2019. The outbreak was constituted as a Public Health Emergency of International Concern (PHEIC) on 30 January 2021 and was declared as a pandemic by the World Health Organization (WHO) on 11 March 2021 [1]. 

One of the most essential strategies to control a pandemic is the rapid development of safe and effective vaccines. The isolation of the virus in 2019 accelerated the development of various vaccines by 2020 [2]. 

The rapid development of COVID-19 vaccines for Severe Acute Respiratory Syndrome Coronavirus 2 (SARS-CoV-2) is a milestone that changed the course of the pandemic, saving tens of millions of lives globally [3]. As of 23 January 2023, a total of 13,156,047,747 COVID-19 vaccine doses have been administered worldwide [4]. 

There are a number of COVID-19 vaccines currently used with proven safety and efficacy. Vaccines are potentially associated with adverse events. Adverse events can present as local or systemic [5]. Though the most frequently reported adverse events following SARS-CoV-2 vaccines are local and systemic adverse events [6], the reporting of allergic reactions, in particular at the beginning of SARS-CoV-2 vaccination with mRNA vaccines, the anaphylaxis and non-anaphylaxis allergic reactions based on spontaneous reports to VAERS, and increased public and health care provider awareness have become so significant for community acceptance of COVID-19 vaccines [7,8].

During the pandemic, both inactivated (CoronaVac) and mRNA (BNT162B2 mRNA vaccines have been used in primary vaccination in Turkey. The inactivated CoronaVac and BNT162B2 mRNA vaccination programs were launched on 13 January 2021 and 2 April 2021, respectively. For individuals who have completed their primary vaccination with inactivated vaccines, the implementation of a booster dose is recommended at the sixth month, as of July 2021.

Turkey started its own COVID-19 vaccine development program in 2020. TURKOVAC™ is a whole-virion inactivated COVID-19 vaccine which was developed and granted emergency use and conditional marketing authorization in December 2021 in Türkiye. Inactivated SARS-CoV-2 virus (hCoV-19/Turkey/ERAGEM-001/2020 strain) vaccine antigen is produced in VERO CCL-81. The vaccine contains aluminum hydroxide (0.5 mg per dose) as adjuvant, phosphate-buffered saline, and water and was manufactured by SBT Science and Biotechnologies. It is available as multi-dose vials, five-dose ready for i.m. injection; the dosage is 3 μg/0.5 mL per injection. In phase I-II trials, two-dose regimens of the vaccine had an acceptable safety and tolerability profile and elicited comparable neutralizing antibody responses and seroconversion rates exceeding 95% [9]. The relative risk reduction with TURKOVAC™ compared to another inactivated SARS-CoV-2 vaccine (CoronaVac) in phase III clinical trial was 41.03% (95% CI 12.95–60.06) for preventing PCR-confirmed symptomatic COVID-19 [10]. The phase IV trial is currently ongoing.

In this article, we present a preliminary analysis of allergy data of our study, in which a large number of (29,584) participants reported adverse events following the use of TURKOVAC™ vaccine as the first, the second, or the booster dose through a telephone survey using a structured questionnaire conducted in 15 provinces in Türkiye.

## 2. Patients/Materials and Methods

In this cohort study, between February and May 2022, in the selected 15 provinces having an adequate number of health care personnel in the community health centers to conduct the study, 32,300 people having the first, the second, or the booster dose of the vaccination as TURKOVAC were invited to the survey. A total of 29,584 people voluntarily agreed to participate to the survey and were given a structured questionnaire after a minimum of 10 days following the vaccination. The response rate was 91.6%.

The inclusion criteria were having been administered the TURKOVAC vaccine as the first or the second dose of the primary vaccination or as the booster dose following two-dose inactivated (CoronaVac) or two-dose mRNA (BNT162B2) vaccines as the primary vaccination protocol and not having COVID-19 (confirmed by PCR) in the last six months. In our survey, prior COVID-19 vaccine doses and vaccine type were not asked.

The distribution of the baseline, such as age group and sex, and allergic reaction characteristics of the participants were presented by using frequency, percentage, mean ± SD, and median (Min–Max).

This study was approved by Ankara City Hospital Ethics Committee No:2. The Ethical Committee approval permits contacting vaccinated people by phone and to conduct questionnaires for safety monitoring. We have no access to any other data regarding previous vaccination(s) and detailed medical history. Verbal consent was also obtained before the telephone survey, at the beginning of the phone conversation.

People were contacted by community health center personnel. Each participant was called (using their landline/mobile phone numbers that they stated) three times; if no response was obtained from the person, he/she was assumed as “no response”.

Regarding the allergic-adverse-event-reporting phase of the questionnaire, participants were asked to answer if they have any history allergy/allergens, the characteristic details of the allergens, and if they experienced any allergic reaction following vaccination and the onset time of the allergic reaction.

All data were analyzed using SPSS software (v ersion-23, IBM Corp., Armonk, NY, USA).

## 3. Results

The baseline characteristics of the participants are presented in Table 1 and Table 2.

In this study, 29,584 people participated in the telephone survey. Of the participants, 54.4% (16,097) were male and 45.6% (13,487) of them were female. Of the participants, 66.4% were under 65 years of age. Only 1.6% of the participants were below 20 years of age. The predominant age group was 65–69 years (11.8%). The percentages of other age groups are as follows: 20–24 (2.8%), 25–29 (4.0%), 30–34 (5.4%), 35–39 (7.4%), 40–44 (8.5%), 45–49 (8.1%), 50–54 (8.4%), 55–59 (9.7%), 60–64 (10.5%), 65–69 (11.8%), 70–74 (8.4%), 75+ (7.8%). Of the participants, 5.6% (1663) had an unknown age. Mean ± SD: 52.9 ± 16.2, median (Min–Max): 54.5 (18–102).

The distribution of self-reported history of allergy and self-reported allergic reaction experience following vaccination by age group is presented in Table 3. The distribution of percentages of self-reported history of allergy and self-reported allergic reaction experience following vaccination by age group have a similar trend by age group. For the participants under 20 years of age, the percentage of self-reported history of allergy was 4.5% (21 people) and the percentage of self-reported allergic reaction experience following vaccination was 0.4% (two people). For the participants in the 20–24 age group, the percentage of self-reported history of allergy was 5.9% (50 people) and the percentage of self-reported allergic reaction experience following vaccination was 0.2% (two people). For the participants in the 25–29 age group, the percentage of self-reported history of allergy was 5.9% (70 people) and the percentage of self-reported allergic reaction experience following vaccination was 0.7% (eight people). For the participants in the 30–34 age group, the percentage of self-reported history of allergy was 4.5% (71 people) and the percentage of self-reported allergic reaction experience following vaccination was 0.4% (seven people). For the participants in the 35–39 age group, the percentage of self-reported history of allergy was 4.9% (107 people) and the percentage of self-reported allergic reaction experience following vaccination was 0.4% (nine people). For the participants in the 40–44 age group, the percentage of self-reported history of allergy was 5.0% (127 people) and the percentage of self-reported allergic reaction experience following vaccination was 0.4% (10 people). For the participants in the 45–49 age group, the percentage of self-reported history of allergy was 4.9% (117 people) and the percentage of self-reported allergic reaction experience following vaccination was 0.5% (12 people). For the participants in the 50–54 age group, the percentage of self-reported history of allergy was 5.9% (135 people) and the percentage of self-reported allergic reaction experience following vaccination was 0.4% (10 people). For the participants in the 55–59 age group, the percentage of self-reported history of allergy was 5.3% (153 people) and the percentage of self-reported allergic reaction experience following vaccination was 0.7% (21 people). For the participants in the 60–64 age group, the percentage of self-reported history of allergy was 4.7% (146 people) and the percentage of self-reported allergic reaction experience following vaccination was 0.4% (11 people). For the participants in the 65–69 age group, the percentage of self-reported history of allergy was 3.4% (118 people) and the percentage of self-reported allergic reaction experience following vaccination was 0.5% (19 people). Among the 70–74 age group participants, the percentage of self-reported history of allergy was 3.3% (83 people) and the percentage of self-reported allergic reaction experience following vaccination was 0.9% (22 people). Of the participants above 75 years of age, the percentage of self-reported history of allergy was 2.7% (61 people) and the percentage of self-reported allergic reaction experience following vaccination was 0.3% (six people). For the participants with unknown age, the percentage of self-reported history of allergy was 3.5% (56 people) and the percentage of self-reported allergic reaction experience following vaccination was 0.9% (three people).

Of the participants, 46.9% (13,869 people) reported to have any adverse reaction following the vaccination.

Only 142 people reported to experience any allergic reaction following vaccination. This number comprises 0.5% of the total our survey participants and 1.0% of the participants reporting any adverse reaction. Of these 142 participants, 55.6% of them (79 people) were male and 44.4% of them (63 people) were female. Of the participants reporting any allergic reaction, 12 of them (8.5%) reported to be given medical treatment and prescription in a health center. No hospitalization was recorded.

Of the participants, 4.4% (1315 people) reported to have a self-reported allergy history. Of the participants who reported to have any self-reported allergy history, 42.7% were male.

The distribution of self-reported allergens by the participants with a known history of allergy is presented in Table 4. The most reported allergens among participants with a known history of allergy were drugs (26.8%) (antibiotics, other medicines, and analgesics), pollen (23.2%), and house dust/dust (18.0%). Other reported allergens/sources were food (12.6%), unknown (9.0%), bees (3.6%), irritants (Detergents, chemicals, metal, wool) (3.3%), allergic rhinitis/eczema (2.6%), perfume (2.1%), grass/other plants (1.8%), sun (1.8%), animals (1.4%), hot/cold/humidity (1.2%), tobacco smoke/other smoke (1.1%), insect bite (0.4%), latex (0.1%), and psychogenic (0.1%).

Among the participants without a known history of allergy (*n* = 28,269), 0.4% of them (110 people) reported to experience an allergic reaction following the vaccination, and 5.4% of the allergic reactions (six people) were reported to be treated in a health center. The percentage of the participants given any medical treatment among the participants without a known history of allergy was 0.02%. Among the participants reported to experience any allergic reaction with a recorded onset of time (*n* = 132), 59.1% of them (78 people) reported to have an allergic reaction on the first day, 10.6% of them (14 people) on the second day, and 6.8% of them (nine people) on the third day and after following the vaccination. Almost all were reported as skin reactions. No immediate or anaphylactic reaction was reported.

A comparison of the experience of any allergic reaction and self-reported history of allergy following the vaccination is given in Figure 1. Among the participants with a known history of allergy (*n* = 1315), 32 people (2.4% of them) reported experiencing an allergic reaction following the vaccination. Among those 32 people, 18.7% of them (six people) reported to be prescribed medical treatment. The percentage of the participants given any medical treatment among the participants with a known history of allergy is 0.4%. The distribution of allergens reported by those six people are: food (two people), pollen (two people), house dust/dust (two people), latex (one person), sun (one person), and unknown (one person).

In regard to the distribution of self-reported post-vaccination allergic reaction rates among participants with a self-reported history of allergy, stratified by age group, the self-reported post-vaccination allergic reaction rates were high among all age groups and significantly higher among <20 (21.2 times), 30–34 (8.6 times), 35–39 (24.4 times), 40–44 (18.8 times), 45–49 (6.5 times), 55–59 (7.1 times), 60–64 (7.6 times), and 65–69 (5.4 times) age groups (Table 5).

## 4. Discussion

The COVID-19 vaccines used are based on different vaccine platforms. The different platforms are based on nucleic acids, artificial vectors or recombinant viruses, virus protein subunits, or live attenuated or inactivated viruses [11]. The Pfizer-BioNTech (BNT162b2) vaccines against COVID-19 are mRNA-based vaccines with the lipid nanoparticle encapsulated, and encode the prefusion-stabilized full-length spike protein of SARS-CoV-2. The Moderna (mRNA-1273) vaccines are also mRNA-based vaccines. The Janssen Ad26.COV2.S vaccine comprises a recombinant, replication-incompetent adenovirus serotype 26 (Ad26) vector, and encodes a stabilized full-length spike protein of SARS-CoV-2 [12]. Randomized controlled trials with these vaccines showed a low rate of severe side effects [12]. Vaccines and their side effects are associated with vaccination acceptance [13,14].

In this study, we conducted a rapid self-assessment of allergic adverse events following the first, second, or the booster doses of the TURKOVAC™ vaccine.

In our study, only 28.0% of the participants were above 65 years of age, although the elderly are the most vulnerable and among the top priority population for COVID-19. Inactivated CoronaVac and BNT162B2 mRNA vaccinations were started on January 2021 and April 2021, respectively. The elderly population was among the primary risk groups when COVID-19 vaccination was launched. Since this group has already been vaccinated with primary and even booster doses of inactivated and mRNA vaccines that are currently in use in Türkiye, the percentage of the participants above 65 years of age who were vaccinated with the TURKOVAC vaccine is relatively low.

In this study, the percentage of the participants under 20 years of age was notably low (1.6%). TURKOVAC vaccination was initiated from 18 years of age onwards.

Of the participants, 4.4% reported to have any allergy history. The most reported allergens were drugs, which constituted 1.2% of the total number of participants and 26.8% of the participants with a known history of allergy. In a phase III trial of TURKOVAC, the most common pre-existing condition among the TURKOVAC arm was allergic conditions, which constituted 2.6% (six persons) of all (456 persons) [10].

In this study, 0.5% of the total participants reported experiencing any allergic reaction following the vaccination. In phase III trial of the vaccine, the incidences of adverse events overall were 58.8% in TURKOVAC and 49.7% in CoronaVac arms (*p* = 0.006), with no fatalities or grade-four adverse events [10]. In a self-administered online survey following inactivated SARS-CoV-2 vaccine (CoronaVac) conducted in China among 1526 health care workers between 24 February and 7 March 2021, 1.0% of the participants reported an allergic reaction [15].

In this study, no immediate or anaphylactic reaction was reported by the participants. Anaphylaxis is a serious, life-threatening allergic reaction [16]. According to the American Academy of Allergy, Asthma & Immunology (AAAAI), the most common anaphylactic reactions are to foods, insect stings, medications, and latex [16]. Anaphylaxis as an adverse event following immunization (AEFI) is uncommon, occurring at a rate of less than 1 per million doses for most vaccines [17].

The incidence of SARS-CoV-2 vaccine anaphylaxis is 7.91 cases per million [18]. The prevalence of COVID-19 mRNA vaccine-associated anaphylaxis is very low and proposed to be associated with polyethylene glycol (PEG) and derivatives. Anaphylaxis rates for Pfizer-BioNTech and Moderna are reported as 11.1 and 2.5 per million doses in U.S. Non-anaphylactic reactions occur at higher rate; however, cutaneous reactions are largely self-limited [11,17,18,19,20,21].

Among the participants reporting to experience any allergic reaction, 55.6% were male (whereas only 42.7% of the participants who reported to have any self-reported allergy history were male). This finding is different from the findings of various studies on allergic adverse events. In a study in which reported allergic adverse events (between December 2020 and June 2021) following COVID-19 vaccinations were analyzed, 84.6% of the patients with allergic reactions were women [10]. In another study on reports of adverse events following two or three doses of the Pfizer-BioNTech COVID-19 vaccine obtained from four cross-sectional studies, the risks of adverse events (including allergic adverse events) were higher following the second dose and consistently higher in females at all ages [17]. There are also studies reporting a female predominance of allergic reactions following administration of other vaccines as well [22]. In another study where China’s adverse events following immunization (AEFI) system data (of 53 vaccines) from 2015 to 2018 were analyzed, the incidence (AEFI) in males was higher than that in females [23].

Besides routine adverse event following immunization (AEFI) monitoring systems, rapid AEFI assessment becomes quite significant for the community acceptance of newly introduced vaccines such as COVID-19 vaccines. Telephone surveys have several advantages, such as providing a geographically widely distributed sample and the inclusion of additional participants living in remote areas, providing a rapid and high population coverage and lower costs of interview [24].

This study has some limitations. In this study, only some factors (age, sex, presence of underlying chronic disease, known history of allergy) could be assessed; the inclusion of the other factors such as vaccine dose, vaccine used for previous vaccination(s), nature of the allergic reaction, and allergic reaction history following the previous COVID-19 vaccination(s) could provide a more detailed evaluation point of view. Self-reporting as the main method used in our survey might have led to biases regarding the validity and severity of the allergic reaction.

## 5. Conclusions

In our study, only 0.5% of the participants reported to experience any self-reported allergic reaction. Male predominance (55.6%) was observed among participants reported to experience any allergic reaction. A known history of allergy increased the risk of having an allergic experience by approximately six times following vaccination. These results present valuable information for the community and may contribute to growing vaccine acceptance.

In conclusion, as a whole-virion inactivated SARS-CoV-2 vaccine, the TURKOVAC™ vaccine, with a low allergic reaction-related adverse event profile, can be an alternative to other COVID-19 vaccines. Larger studies may provide more detailed data.

## Figures and Tables

**Figure 1 vaccines-11-00437-f001:**
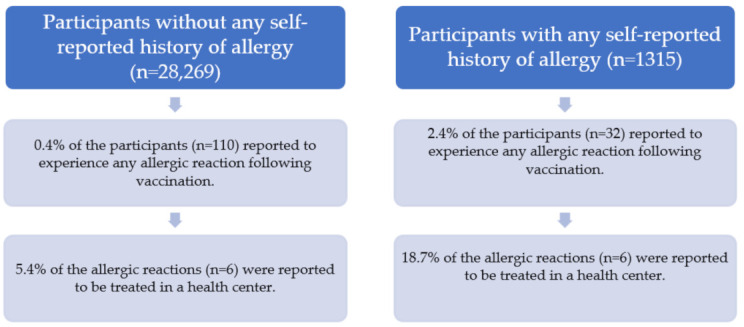
Comparison of experience of any allergic reaction and self-reported history of allergy following the vaccination.

**Table 1 vaccines-11-00437-t001:** Distribution of baseline characteristics of the participants.

	(*n*)	(%)
**Sex**		
Male	16,097	54.4
Female	13,487	45.6
**Age Group**		
<20	466	1.6
20–24	841	2.8
25–29	1189	4.0
30–34	1590	5.4
35–39	2192	7.4
40–44	2520	8.5
45–49	2394	8.1
50–54	2481	8.4
55–59	2876	9.7
60–64	3099	10.5
65–69	3496	11.8
70–74	2483	8.4
75+	2294	7.8
Unknown	1663	5.6

**Table 2 vaccines-11-00437-t002:** Distribution of mean and median age of the participants by sex.

Sex	Mean ± SD	Med (Min–Max)
Male	53.2 ± 15.9	54.5 (18–100)
Female	52.5 ± 16.7	54.5 (18–102)
Total	52.9 ± 16.2	54.5 (18–102)

**Table 3 vaccines-11-00437-t003:** Distribution of self-reported history of allergy and self-reported allergic reaction rates following vaccination by age group.

Age Group	Self-Reported History of Allergy (*n* = 1315)		Allergic Reaction Following Vaccination (*n* = 142)	
	#	Rate %	#	Rate %
<20	21	4.5	2	0.4
20–24	50	5.9	2	0.2
25–29	70	5.9	8	0.7
30–34	71	4.5	7	0.4
35–39	107	4.9	9	0.4
40–44	127	5.0	10	0.4
45–49	117	4.9	12	0.5
50–54	135	5.9	10	0.4
55–59	153	5.3	21	0.7
60–64	146	4.7	11	0.4
65–69	118	3.4	19	0.5
70–74	83	3.3	22	0.9
75+	61	2.7	6	0.3
Unknown	56	3.5	3	0.9

**Table 4 vaccines-11-00437-t004:** Distribution of self-reported allergens/sources by the participants with known history of allergy.

Allergen	#	% Among Total Participants (*n* = 29,584)	% Among Participants with Known History of Allergy (*n* = 1315)
Drugs	352	1.190	26.768
Pollen	305	1.031	23.193
House dust/dust	237	0.801	18.022
Food	166	0.561	12.623
Unknown	118	0.399	8.973
Bees	48	0.162	3.650
Irritants (detergents, chemicals, metal, wool)	43	0.145	3.269
Allergic rhinitis/eczema	34	0.115	2.585
Perfume	28	0.095	2.129
Grass/other plants	24	0.081	1.825
Sun	24	0.081	1.825
Animals	18	0.061	1.368
Hot/cold/humidity	16	0.054	1.216
Tobacco smoke/other smoke	15	0.051	1.140
Insect bite	5	0.017	0.380
Latex	2	0.007	0.152
Psychogenic	1	0.003	0.076

**Table 5 vaccines-11-00437-t005:** Distribution of self-reported post-vaccination allergic reaction rates among participants with self-reported history of allergy, stratified by age group.

	Self-Reported Post-Vaccination Allergic Reaction Rates				RR (95% CI)	*p*
	Participants with Self-Reported History of Allergy		Participants without Self-Reported History of Allergy			
Age Group	(*n*)	Rate %	(*n*)	Rate %		
<20	1	4.8	1	0.2	21.2 (1.4–327.2)	0.002
20–24	-	-	2	0.3	-	-
25–29	1	1.4	7	0.6	2.3 (0.3–18.3)	0.385
30–34	2	2.8	5	0.3	8.6 (1.7–43.4)	0.036
35–39	5	4.7	4	0.2	24.4 (6.6–89.4)	<0.001
40–44	5	3.9	5	0.2	18.8 (5.5–64.3)	<0.001
45–49	3	2.6	9	0.4	6.5 (1.8–23.7)	0.018
50–54	1	0.7	9	0.4	1.9 (0.3–15.1)	0.429
55–59	6	3.9	15	0.6	7.1 (2.8–18.1)	0.001
60–64	3	2.1	8	0.3	7.6 (2.0–28.3)	0.013
65–69	3	2.5	16	0.5	5.4 (1.6–18.2)	0.024
70–74	2	2.4	20	0.8	2.9 (0.7–12.2)	1.660
75>	-	-	6	0.3	-	1.000

## Data Availability

Data supporting the findings of this study are available on request from the corresponding author, A.K. The data are not publicly available due to their containing information that could compromise the privacy of research participants.

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
