# Peer review of "Self-Reported Allergic Adverse Events Following Inactivated SARS-CoV-2 Vaccine (TURKOVAC™) among General and High-Risk Population"

_vaccines, 2023, doi:10.3390/vaccines11020437_

Round 1
Reviewer 1 Report
1. The authors indicated there are total 29,584 voluntary participants, male 16,097 and female 13,487. The manuscript also indicated that the participants without known history of allergy is 29,442, and those with known history of allergy is 1,320. The combination of these two groups therefore is 30,762 (29,442+1320). This number is not match with the total number of 29,584.
2. Check table 1, the line Age Group “466, 1.6.”
3. Table 1a included some information also presented in table 1b. Suggest to combine table 1a and table 1b or remove the repeated information in table 1a.
4. Line 116-118: “Following the vaccination, only 0.5% of the total participants and 1.0% of the participants reporting any adverse reaction (142 people) reporting to experience any allergic reaction.” This sentence is not clear and need to rewrite.
5. In line 118 (as well as in several places in the entire manuscript): “Fifty-five-point six percent…”. It may be more suitable in bio-medical research field to present it as “55.6%”.
Author Response
Manuscript ID: vaccines-2136577
Type of manuscript: Article
Title: Self-reported allergic adverse events following inactivated SARS-CoV-2 Vaccine (TURKOVAC™) among general and high-risk population
Reviewer I
First of all, we would like to thank you for your kind recommendation to improve our paper,
- The authors indicated there are total 29,584 voluntary participants, male 16,097 and female 13,487. The manuscript also indicated that the participants without known history of allergy is 29,442, and those with known history of allergy is 1,320. The combination of these two groups therefore is 30,762 (29,442+1320). This number is not match with the total number of 29,584.
The number of the participants without known history of allergy was corrected as 28,269, and those with known history of allergy was corrected as 1,315. Relevant percentages were also corrected in Table 2 and in Figure 1.
- Check table 1, the line Age Group “466, 1.6.”
Table 1 was checked and the shift was corrected in the table and relevant figures were corrected in the manuscript.
- Table 1a included some information also presented in table 1b. Suggest to combine table 1a and table 1b or remove the repeated information in table 1a.
The repeated information in Table 1a was removed.
- Line 116-118: “Following the vaccination, only 0.5% of the total participants and 1.0% of the participants reporting any adverse reaction (142 people) reporting to experience any allergic reaction.” This sentence is not clear and need to rewrite.
The sentence was checked and was rewritten as “Only 142 people reported to experience any allergic reaction following vaccination. This number comprises 0.5% of the total participants and 1.0% of the participants reporting any adverse reaction”.
- In line 118 (as well as in several places in the entire manuscript): “Fifty-five-point six percent…”. It may be more suitable in bio-medical research field to present it as “55.6%”.
All recommended corrections were made and revised as figures.
Thank you for your kind review.

Reviewer 2 Report
The work by Kara et al., show a preliminary analysis of allergic adverse events following the use of TURKOVAC™ vaccine after the first, the second or the booster dose using a questionnaire conducted in 15 provinces in Türkiye via telephone survey with of 29,584 participants.
The authors report that only 0.5% of the participants experienced any allergic reaction and 8.5% of them received a medical treatment in a health center. No hospitalization was recorded. 4.5% of the participants reported to be allergic. Among the participants without a known history of allergy, 0.4% of them reported to experience an allergic reaction following the vaccination. No immediate or anaphylactic reaction was reported. Among the participants with known history of allergy, 2.4% of them reported to experience any allergic reaction following the vaccination.
The authors conclude that although the incidence of allergic reactions in their study was low, a previous history of allergy increased by 6 times the risk of having an allergic experience following vaccination. They suggest that the TURKOVAC™ vaccine has a low allergic reaction-related adverse event profile and could be an alternative to other COVID-19 vaccines.
The major drawback of the Ms is that the conclusion is not supported by the data shown since they did not compare in their study with other COVID-19 vaccines. Moreover, they do not discuss the reasons why TURKOVAC™ vaccine was found to be of low allergenicity in spite of alum being an allergy-prone adjuvant?
Author Response
Manuscript ID: vaccines-2136577
Type of manuscript: Article
Title: Self-reported allergic adverse events following inactivated SARS-CoV-2 Vaccine (TURKOVAC™) among general and high-risk population
Reviewer II
First of all, we would like to thank you for your kind recommendation to improve our paper,
The work by Kara et al., show a preliminary analysis of allergic adverse events following the use of TURKOVAC™ vaccine after the first, the second or the booster dose using a questionnaire conducted in 15 provinces in Türkiye via telephone survey with of 29,584 participants.
The authors report that only 0.5% of the participants experienced any allergic reaction and 8.5% of them received a medical treatment in a health center. No hospitalization was recorded. 4.5% of the participants reported to be allergic. Among the participants without a known history of allergy, 0.4% of them reported to experience an allergic reaction following the vaccination. No immediate or anaphylactic reaction was reported. Among the participants with known history of allergy, 2.4% of them reported to experience any allergic reaction following the vaccination.
The authors conclude that although the incidence of allergic reactions in their study was low, a previous history of allergy increased by 6 times the risk of having an allergic experience following vaccination. They suggest that the TURKOVAC™ vaccine has a low allergic reaction-related adverse event profile and could be an alternative to other COVID-19 vaccines.
The major drawback of the Ms is that the conclusion is not supported by the data shown since they did not compare in their study with other COVID-19 vaccines. Moreover, they do not discuss the reasons why TURKOVAC™ vaccine was found to be of low allergenicity in spite of alum being an allergy-prone adjuvant?
Discussion was improved to include comparisons with other COVID-19 vaccines. There is no increased allergic reaction with vaccines containing aluminium as adjuvant compared to the vaccines containing other substances as adjuvant. Thank you for kind review.

Round 2
Reviewer 1 Report
No further comment.
Reviewer 2 Report
In the revised form the authors took into account my main objection and accommodated the ms to sound accordingly.